# Impact of Vitamin B12 Insufficiency on Sarcopenia in Community-Dwelling Older Korean Adults

**DOI:** 10.3390/ijerph182312433

**Published:** 2021-11-26

**Authors:** Seon A Chae, Hee-Sang Kim, Jong Ha Lee, Dong Hwan Yun, Jinmann Chon, Myung Chul Yoo, Yeocheon Yun, Seung Don Yoo, Dong Hwan Kim, Seung Ah Lee, Sung Joon Chung, Yunsoo Soh, Chang Won Won

**Affiliations:** 1Department of Physical Medicine and Rehabilitation Medicine, Kyung Hee University Medical Center, Seoul 02447, Korea; chae14827@daum.net (S.A.C.); kimhsmd@khu.ac.kr (H.-S.K.); lukaslee@hanmail.net (J.H.L.); dhyun@khu.ac.kr (D.H.Y.); kkangmann@naver.com (J.C.); famousir@naver.com (M.C.Y.); yunsn123@naver.com (Y.Y.); 2Department of Physical Medicine and Rehabilitation, Kyung Hee University Hospital at Gangdong, Seoul 05278, Korea; kidlife@khu.ac.kr (S.D.Y.); kdhkjr@hanmail.net (D.H.K.); lsarang80@gmail.com (S.A.L.); sungjoon.chung@gmail.com (S.J.C.); 3Department of Family Medicine, Kyung Hee University Medical Center, Seoul 02447, Korea

**Keywords:** vitamin B12, sarcopenia, aging, walking speed, hand grip strength

## Abstract

Vitamin B12 (B12) is involved as a cofactor in the synthesis of myelin. A lack of B12 impairs peripheral nerve production, which can contribute to sarcopenia. In this cross-sectional study, we aimed to investigate the relationship between B12 insufficiency and sarcopenia in community-dwelling older Korean adults. A total of 2325 (1112 men; 1213 women) adults aged 70–84 years were recruited. The tools used for sarcopenia were based on the Asian Working Group for Sarcopenia (AWGS) guidelines. Individuals with low appendicular skeletal muscle mass index (ASMI) (<7.0 kg/m^2^ for men; <5.4 kg/m^2^ for women) and low hand grip strength (HGS) (<28 kg for men; <18 kg for women) were defined as the sarcopenia group. Among this group, those who showed low physical performance (≤9 points on the Short Physical Performance Battery (SPPB)) were defined as the severe sarcopenia group. B12 concentrations were classified into insufficient (<350 pg/mL) and sufficient (≥350 pg/mL). Univariate and multivariate logistic regression analyses were used to evaluate the relationship between sarcopenia and B12 levels. Low ASMI showed a high incidence in the B12-insufficient group. However, HGS, SPPB, and the severity of sarcopenia showed no correlation with B12. Further, insufficient B12 may affect muscle quantity rather than muscle strength or physical performance.

## 1. Introduction

Vitamin B12 (B12), also called cobalamin, is a water-soluble B vitamin that is required as a cofactor in DNA synthesis in both fatty acid and amino acid metabolism in the human body [1]. B12 also helps turn homocysteine into methionine and plays an important role in the synthesis of myelin in the nervous system [2]. B12 deficiency has a generally low incidence because the body requires small amounts of B12 [3]. However, this deficiency is still likely to occur in vegetarians, absorptive disorder patients after gastrectomy, individuals using long-term antacid therapy including proton pump inhibitors or H2 blockers, chronic alcoholics, and the elderly [3]. Although the absorption of B12 itself does not decrease in most elderly individuals, the secretion of pepsin and gastric juice decreases with age, and atrophic gastritis occurs more frequently. As a result, the absorption rate of B12 is low [4,5]. Demyelinating peripheral nerve disorders observed in the elderly with B12 deficiency with significant nerve fiber damage often cause muscle weakness, numbness, and pain in the distal limbs, resulting in impaired balance, gait ataxia, and even sarcopenia [6,7,8].

Sarcopenia occurs when the age-related loss of skeletal muscle mass is accompanied by muscle strength loss or physical dysfunction. This condition is responsible for disability and even increases mortality [9,10]. The prevalence of sarcopenia in Asia differs depending on the diagnostic method and sex. Sarcopenia occurs in approximately 10% of the population over 65 years of age and increases with age, with the highest incidence of more than 50% after the age of 80 [11,12]. In 2010, the European Working Group on Sarcopenia in Older People proposed a diagnostic algorithm for sarcopenia. Due to the relatively smaller body size, higher adiposity, less mechanization, and more physical activity of Asians, in 2014 the Asian Working Group for Sarcopenia (AWGS) proposed a diagnostic algorithm based on Asian data and revised it in 2019 [13]. Factors related to sarcopenia in the elderly include an imbalance between protein breakdown and synthesis, inflammatory cytokines, cortisol, sex hormones, and lifestyle-related factors such as nutritional intake and physical activity [14,15,16]. Furthermore, age-related conditions such as loss of muscle innervation and reduced myonuclei replacement by satellite cells could cause the loss of muscle and function; these factors might co-lead to sarcopenia development [17]. Previous studies have reported that B12 deficiency may be associated with sarcopenia in the elderly population [6]. However, in many of these studies the analysis of sarcopenia was limited to measuring physical functions such as hand grip strength (HGS) and walking speed rather than measuring the amount of skeletal muscle mass. Therefore, it was difficult to accurately evaluate the reduction in lean skeletal muscle mass.

The purpose of this cross-sectional study was to examine the effect of B12 levels on sarcopenia, including skeletal muscle mass, using dual-energy X-ray absorptiometry (DEXA) and physical performance in community-dwelling older adults. For the study, we investigated baseline data from the Korean Frailty and Aging Cohort Study (KFACS).

## 2. Materials and Methods

### 2.1. Data and Study Population

We used data from the KFACS to investigate the relationship between B12 levels and sarcopenia as defined by the AWGS. The KFACS is a nationwide, multicenter study performed in eight medical and two public health centers across South Korea of older community dwellers between 70 and 84 years of age. Among the 3014 participants, data were excluded if there were incomplete data on B12 levels or DEXA, weakness due to cerebrovascular accident, deformity or motor deficit in the extremities, blindness, or an inability to complete the physical performance test including a 4 m walking speed and Short Physical Performance Battery (SPPB) test. Finally, 2325 participants (1112 men; 1213 women) were included in the study (Figure 1). Baseline demographic data and medical history, including age, sex, years of education (<6, 6–12, and >12 years), location of residence (rural or urban), smoking and alcohol habits, body mass index (BMI), and chronic comorbidities, were collected. Those who smoked more than one cigarette per week and drank alcohol at least once a week were defined as smokers and drinkers, respectively. The KFACS protocol was approved by the Institutional Review Board (IRB) of the Clinical Research Ethics Committee of the Kyung Hee University Medical Center (IRB number: 2015-12-103), and all participants provided written informed consent.

### 2.2. Vitamin B12

B12 from serum samples was measured using the Architect Vitamin Kit (Abbott Diagnostics, Lake Forest, IL, USA). In a previous study, the participants were divided by serum B12 concentrations > 350 pg/mL, which was found to be the protective level of myelin synthesis in the nervous system. Individuals with a serum B12 concentration of <350 pg/mL were diagnosed with insufficient B12 levels [5,18]. We divided B12 concentrations into clinically relevant categories: insufficiency group (<350 pg/mL, same as <258.3 pmol/L) and sufficiency group (≥350 pg/mL, ≥258.3 pmol/L).

### 2.3. Sarcopenia

Sarcopenia was defined using the AWGS diagnostic criteria, which was updated in 2019 [13]. Participants with low muscle mass plus either low muscle strength or low physical performance were included in the sarcopenia group. Participants in the sarcopenia group with low muscle mass plus both low muscle strength and low physical performance were further categorized into the severe sarcopenia group.

(1)Muscle strength: HGS was measured using a hand dynamometer (JAMAR, Bolingbrook, IL, USA). HGS was measured twice on both sides, with the elbow flexed at 90° in a sitting position, and the highest value was obtained (cut-off value: <28 kg for men; <18 kg for women).(2)Appendicular skeletal muscle mass (ASM): Among the variety of techniques that can estimate muscle quantity, DEXA is more widely available and useful for its non-invasiveness, accuracy, and convenience for determining muscle quantity [19]. The appendicular skeletal muscle mass index (ASMI) was compared at different heights using height squared (ASM/height^2^). The cut-off values for sarcopenia were <7.0 kg/m^2^ for men and <5.4 kg/m^2^ for women.(3)Physical performance was measured using the SPPB. The SPPB is a well-established performance test that evaluates the physical function of the lower extremities in older adults. It comprises the standing balance test, walking speed, and five repeated chair stands. Each of these components is scored on a 4-point scale, with a higher score indicating better lower extremity physical function. In AWGS, a SPPB score of ≤9 points indicates low physical performance.

### 2.4. Statistical Analysis

The demographic characteristics of the participants based on B12 levels were analyzed using a *t*-test for continuous variables and a chi-square test for categorical variables. The results were expressed as mean ± standard deviation (SD) or number and ratio (%) according to the characteristics of the variables. Univariate and multivariate analyses were performed using logistic regression models to examine the odds ratio (OR) of sarcopenia scales depending on B12 levels. Each multivariate model for multiple interrelations between physical functions and other potential confounding variables such as age, sex, depression, osteoarthritis, osteoporosis, diabetes mellitus, hypertension, smoking, alcohol consumption, location of residence, and BMI was fully adjusted. The collected data were analyzed using SPSS (version 23.0; IBM, Inc., Chicago, IL, USA) software. A *p* value of <0.05 was considered significant.

## 3. Results

The baseline characteristics of the KFACS participants were evaluated based on their B12 levels (Table 1). Of the 2325 participants, 2044 (79.5%) showed B12 sufficiency (≥350 pg/mL), while 281 (20.5%) were classified as having insufficient B12 levels. The mean ages were 77.2 (SD = 3.9) and 76.3 (SD = 3.8) years for the sufficient and insufficient groups, respectively, with a significantly high mean age and a predominance of males in the B12 insufficient group. BMI, years of education, marital status, income, and location of residence were not significantly different between the two groups. In terms of chronic comorbidities, the B12-insufficient group had a higher proportion of individuals with diabetes mellitus.

The results of sarcopenia parameters including ASMI, HGS, and SPPB of the two groups are shown in Table 2. No statistical significance was observed in the ASMI, HGS, and SPPB parameters. In addition, the incidence of sarcopenia and severe sarcopenia was not significantly different between the two groups.

Table 3 shows the univariate and multivariate logistic regression analyses of sarcopenia parameters according to B12 levels: insufficiency group (<350 pg/mL) and sufficiency (≥350 pg/mL). Sarcopenia and severe sarcopenia defined by the AWGS showed no correlation in the univariate and multivariate logistic regression models. However, in the unadjusted model, the incidence of low ASMI (<7.0 kg/m^2^ for men and <5.4 kg/m^2^ for women) was high in the B12-insufficient group (OR = 1.596, 95% confidence interval (CI) = 1.242–2.051). Moreover, in the model fully adjusted for age, sex, depression, osteoarthritis, osteoporosis, diabetes mellitus, hypertension, smoking, alcohol consumption, location of residence, and BMI, the incidence of low ASMI was significantly higher in the B12-insufficient group (OR = 1.744, CI = 1.301–2.339). The other parameter results were not significant in the univariate and multivariate logistic models.

Table 4 shows the univariate and multivariate logistic regression analysis according other clinically relevant B12 level categories: insufficiency group (<400 pg/mL, same as <295.1 pmol/L) and sufficiency (≥400 pg/mL, ≥295.1 pmol/L). This result also showed similar results, with cuff-off values of 350 pg/mL.

## 4. Discussion

In this study, B12 insufficiency was associated with low muscle mass; the association was significant even after adjusting for confounding factors. Our study measured the amount of ASMI using DEXA according to B12 levels. The ASMI values of <7.0 kg/m^2^ for men and <5.4 kg/m^2^ for women showed a high OR in the B12 insufficiency group; however, the difference was not related to the incidence of sarcopenia or physical performance function.

The relationship between the decrease in muscle mass and B12 found in this study can be explained by the following hypothesis: B12 is involved in DNA synthesis and the metabolic process of cell growth in the human body by acting as a cofactor for methionine synthase and L-methyl-malonyl-coenzyme A mutase [1]. Muscle mass might be reduced in the elderly with B12 insufficiency because the proliferation and differentiation of satellite cells, stem cells necessary for skeletal muscle regeneration, and providing myonuclei to preexisting muscle fibers would be decreased [17,20]. B12 also plays an essential role in the development, myelination, and maintenance of both the central and peripheral nervous systems [21]. Since B12 affects nerve myelination, B12 insufficiency causes demyelination, which slows down messages sent along axons and causes axonal deterioration and a decrease in muscle mass [22,23]. Additionally, hyperhomocysteinemia, which is recognized as a sensitive marker for B12 deficiency, has been reported to damage skeletal muscles, as evidenced by elevated muscle-specific creatinine phosphokinase [24]. Hyperhomocysteinemia has also been shown to deteriorate to vascular inflammation, thrombosis, and thromboembolism, resulting in peripheral arterial disease. This arterial damage consequently results in damage, inflammation, and loss of muscle regeneration capability [25]. This supports the results of our study on muscle mass reduction and B12 insufficiency.

A lack of B12 impairs peripheral nerve production, which can lead to a loss of muscle mass. However, because sarcopenia is defined as loss of muscle mass and decreased muscle strength, the direct relationship between B12 and sarcopenia remains controversial. In a study by Tao et al. with 427 hospitalized older adults aged >80 years, no association was found between serum B12 levels and sarcopenia [26]. Although this is consistent with our results, their results were limited because only muscle strength (HGS < 27 kg in men or <16 kg in women) and muscle quantity (calf circumference (CC) of <31 cm) were evaluated for diagnosing sarcopenia. Moreover, CC is useful for screening or detecting sarcopenia, but it is less accurate than ASM determined by DEXA in measuring muscle quantity [26]. Another study of 731 community-dwelling adults aged ≥65 years in Taiwan identified factors associated with sarcopenia, especially nutritional markers. Vitamin D was positively correlated with ASM in men with sarcopenia, whereas there was no correlation between nutritional factors, including B12, and muscle mass. This is consistent with the results of our study; however, the cited study was limited in that the size of the cohort was relatively small, with only 50 sarcopenic older adults [27]. In the KFACS study, we reported that B12 insufficiency was not related to SARC-F, a screening tool for sarcopenia [28]. Because SARC-F is a self-reporting questionnaire, there is a lack of information on muscle mass and low physical performance. In another prospective study of 403 elderly individuals aged >60 years, lean body mass, total skeletal mass, and skeletal muscle mass index were lower in the B12-insufficient group than in the B12-sufficient group [6]. These results support our findings. Our study used DEXA, a method for measuring muscle mass that is more accurate than many methods used in previous studies, and the results showed a significant correlation with B12 insufficiency.

Several previous studies have reported that B12 levels may be associated with physical function. In a cross-sectional study of 703 community-dwelling Caucasian elderly women, low B12 levels were associated with a higher risk of frailty in HGS, endurance, physical activity, and walking speed [29]. Another study of 3015 elderly individuals in the National Health and Nutrition Examination Surveys showed that participants with B12 insufficiency (<350 pg/mL) showed a greater risk of having a disability in activities of daily living and mobility [30]. However, other studies have reported that physical performance does not correlate with B12 levels. In a prospective cohort study of 796 elderly Singaporeans for the association of homocysteine, folate, and B12 with completed performance-oriented mobility assessment (POMA) of gait, balance, and self-reports of instrumental activities of daily living (IADL), the results showed that serum B12 level was not significantly associated with POMA balance, POMA gait, or IADL scores [31]. Another study of 1352 community-dwelling older adults in Amsterdam evaluated the correlation between B12 and physical performance, including walking test, chair stand test, and tandem stand. However, the association was not clearly established [32]. Likewise, in the current study, B12 insufficiency did not increase the risk of muscle weakness or poor physical performance. ASMI, representing muscle mass, is a relatively static and uniform measurement tool, whereas HGS, walking speed, and SPPB evaluate more dynamic and complex physical activities. Furthermore, a patient’s conditions, such as fatigue, mood, or cognitive functions, may affect these physical activities. Therefore, it seems that more variables need to be considered to understand the correlation between B12 levels and physical performance in future studies.

This study had several limitations. First, this was a cross-sectional study. However, the results of this large cohort study with more than 2000 participants are meaningful because of the paucity of previous studies examining the relationship between B12 levels and sarcopenia. Further prospective studies or large randomized controlled trials are needed to confirm our findings. Second, patients’ B12 supplements were not considered in this study. As B12 is included in commercial multivitamin supplements, it may be helpful to investigate this intake in participants. Third, because this study was conducted in older adults, the patients’ fatigue status or body conditions when measuring physical function may have affected the results. Fourth, hematocrit and mean corpuscular volume results were not included in this study. We could not confirm the macrocytic anemia result of myelin synthesis dysfunction due to vitamin B12 deficiency. Finally, B12 insufficiency was determined by serum B12 levels without considering other vitamin B groups (methylmalonic acid and homocysteine) that may affect the biochemical action of B12. Hence, further studies are required to evaluate these correlations.

This study was a large cross-sectional cohort study of the Korean elderly population using baseline data from the KFACS, which is a multicenter, large population cohort study. The tools used for assessing muscle mass, strength, and function were based on the most recent and widely accepted AWGS guidelines for sarcopenia. Our study used DEXA, a method for measuring muscle mass that is more accurate than other measures used in previous studies, and showed a significant correlation with B12 insufficiency. In addition, it is meaningful to statistically control for potential confounders. The definition of sarcopenia defined by AWGS includes both low muscle mass and low muscle strength; however, sarcopenia showed no correlation with B12 levels. These results suggest that insufficient B12 levels affect muscle quantity rather than muscle strength, physical performance, or sarcopenia.

## 5. Conclusions

The results of this cross-sectional study imply that, in older adults, low B12 levels might increase the incidence of low muscle mass, but not sarcopenia. The results were significant even after considering confounding factors including age, sex, depression, osteoarthritis, osteoporosis, diabetes mellitus, hypertension, smoking, alcohol consumption, location of residence, and BMI.

## Figures and Tables

**Figure 1 ijerph-18-12433-f001:**
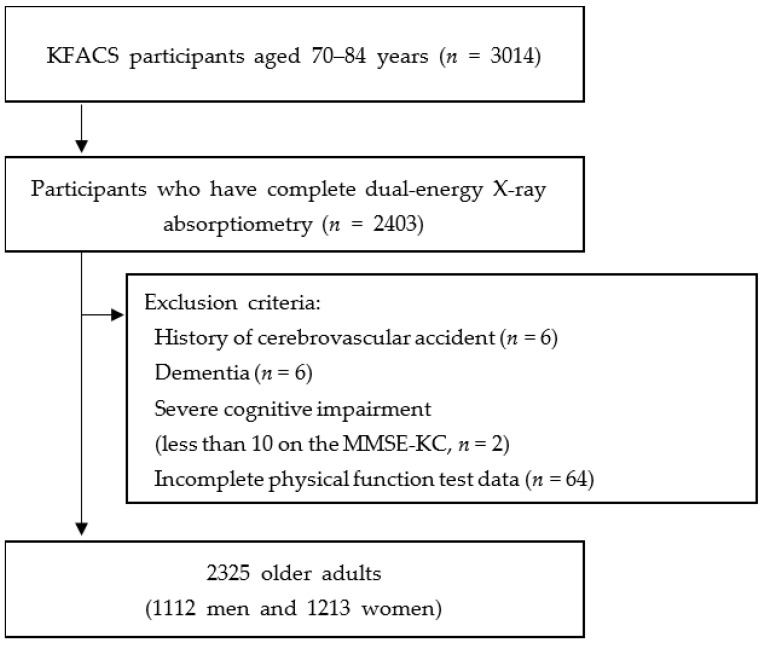
Study flow chart of the participant recruitment process. Abbreviations: KFACS, Korean Frailty and Aging Cohort Study; MMSE-KC, Mini-Mental Status Examination in the Korean version of the Consortium to Establish a Registry for Alzheimer’s Disease (CERAD) assessment packet.

**Table 1 ijerph-18-12433-t001:** Baseline characteristics of participants by vitamin B12 level.

Characteristic		B12 (pg/mL)		Total	*p*
		Sufficiency (≥350, *n* = 2044)	Insufficiency (<350, *n* = 281)	(*n* = 2325)	
Age, mean (SD)		76.3 (3.8)	77.2 (3.9)	76.4 (3.8)	<0.001 *
Sex (*n*, %)	Male	950 (46.5)	162 (57.7)	1112 (47.8)	<0.001 *
	Female	1094 (53.5)	119 (42.3)	1213 (52.2)	
BMI (SD)		24.3 (2.9)	24.5 (3.1)	24.6 (2.9)	0.165
Education years (*n*, %)	Less than 6	853 (41.7)	124 (44.1)	977 (42.0)	0.168
	7–12	765 (37.4)	112 (39.9)	877 (37.7)	
	Over 13 y	426 (20.8)	45 (16.0)	471 (20.3)	
Marriage (*n*, %)	Married	1581 (77.3)	228 (81.1)	1809 (77.8)	0.152
	Not married	463 (22.7)	53 (18.9)	516 (22.2)	
Income per month (Korean million Won ^†^, %)	More than 3	390 (19.1)	51 (18.1)	441 (19.0)	0.781
	1–3	883 (43.2)	118 (42.0)	1001 (43.1)	
	Less than 1	771 (37.7)	112 (39.9)	883 (38.0)	
Residency (*n*, %)	Urban	1667 (81.6)	230 (81.9)	1897 (81.6)	0.905
	Rural	377 (18.4)	51 (18.1)	428 (18.4)	
Current smoker (*n*, %)		655 (32.0)	97 (34.5)	752 (32.3)	0.406
Alcohol use (*n*, %)		1191 (58.3)	175 (62.3)	1366 (58.8)	0.2
Hypertension (*n*, %)		1152 (56.4)	159 (56.6)	1311 (56.4)	0.944
Dyslipidemia (*n*, %)		684 (33.5)	91 (32.4)	775 (33.3)	0.719
Diabetes mellitus (*n*, %)		433 (21.2)	85 (30.2)	518 (22.3)	0.001
Depression (*n*, %)		59 (2.9)	6 (2.1)	65 (2.8)	0.474
OA (*n*, %)		444 (21.7)	69 (24.6)	513 (22.1)	0.283
Osteoporosis (*n*, %)		308 (15.1)	38 (13.5)	346 (14.9)	0.495
WBC (10^3^/μL, SD)		5.8 (1.5)	5.8 (1.5)	5.8 (1.6)	0.768
25(OH)D (ng/mL, SD)		24.0 (10.1)	20.9 (9.3)	23.4 (10.0)	0.056
Hb (g/dL, SD)		13.4 (1.3)	13.4 (1.4)	13.4 (1.2)	0.145
MMSE-KC (SD)		25.8 (3.0)	25.5 (3.5)	25.7 (3.3)	0.023 *

Abbreviations: B12, vitamin b12; BMI, body mass index; OA, osteoarthritis; WBC, white blood cell; 25(OH)D, 25-hydroxyvitamin D; Hb, hemoglobin; MMSE-KC, Mini-Mental Status Examination in the Korean version of the CERAD assessment packet. ^†^ 1 million Korean won = approximately 900 USD, * *p* < 0.05.

**Table 2 ijerph-18-12433-t002:** Sarcopenia parameters of participants by vitamin B12 level.

Characteristic	B12 (pg/mL)		*p*
	Sufficiency (≥350, *n* = 2044)	Insufficiency (<350, *n* = 281)	
HGS (kg, SD)	26.32 (7.4)	27.18 (7.42)	0.066
ASMI (kg/m^2^, SD)	6.44 (0.99)	6.44 (0.96)	0.133
SPPB (SD)	10.84 (1.46)	10.73 (1.48)	0.455
Sarcopenia ^†^ (*n*, %)	225(11.0)	36(12.8)	0.369
Severe Sarcopenia ^§^ (*n*, %)	71(3.5)	12(4.3)	0.5

Abbreviations: HGS, hand grip strength; ASMI, appendicular skeletal muscle mass index; SPPB, short physical performance battery. ^†^ Sarcopenia: Low HGS (<28 kg for men and <18 kg for women) and low ASMI (<7.0 kg/m^2^ for men and <5.4 kg/m^2^ for women). ^§^ Severe sarcopenia; all sarcopenia criteria and low physical performance (SPPB ≤ 9 for both sexes). *p* < 0.05.

**Table 3 ijerph-18-12433-t003:** Logistic regression analysis of sarcopenia definition and parameters by vitamin B12 level: Insufficiency group (<350 pg/mL) and sufficiency (≥350 pg/mL).

	Unadjusted Model		Fully Adjusted Model	
	OR (95% CI)	*p*	OR (95% CI)	*p*
Muscle strength				
Low HGS ^†^	0.987	0.932	0.816	0.213
	(0.728–1.338)		(0.592–1.124)	
Muscle mass				
Low ASMI ^†^	1.596	<0.001 *	1.744	<0.001 *
	(1.242–2.051)		(1.301–2.339)	
Physical performance				
Low SPPB ^†^	1.182	0.308	1.088	0.634
	(0.857–1.629)		(0.769–1.538)	
Sarcopenia ^††^	1.188	0.37	0.991	0.965
	(0.815–1.731)		(0.659–1.489)	
Severe sarcopenia ^§^	1.24	0.500	1.038	0.911
	(0.664–2.316)		(0.540–1.996)	

Abbreviations: OR, odds ratio; CI, confidence interval; HGS, hand grip strength; ASMI, appendicular skeletal muscle mass index; SPPB, short physical performance battery. ^†^ Low HGS (<28 kg for men and <18 kg for women); Low ASMI, <7.0 kg/m for men and <5.4 kg/m for women; Low SPPB ≤ 9 for both sexes; ^††^ Sarcopenia: low HGS and low ASMI. ^§^ Severe sarcopenia, low HGS, low ASMI, and low SPPB. The fully adjusted model was adjusted for age, sex, depression, osteoarthritis, osteoporosis, diabetes mellitus, hypertension, smoking, alcohol consumption, location of residence, and body mass index. * *p* < 0.05.

**Table 4 ijerph-18-12433-t004:** Logistic regression analysis of sarcopenia definition and parameters by vitamin B12 level: Insufficiency group (<400 pg/mL) and sufficiency (≥400 pg/mL).

	Unadjusted Model		Fully Adjusted Model	
	OR (95% CI)	*p*	OR (95% CI)	*p*
Muscle strength				
Low HGS ^†^	1.104	0.434	0.906	0.462
	(0.862–1.416)		(0.696–1.179)	
Muscle mass				
Low ASMI ^†^	1.339	0.006 *	1.478	0.002 *
	(1.086–1.652)		(1.155–1.891)	
Physical performance				
Low SPPB ^†^	1.27	0.077	1.161	0.308
	(0.975–1.655)		(0.871–1.548)	
Sarcopenia ^††^	1.176	0.313	0.975	0.885
	(0.858–1.612)		(0.691–1.375)	
Severe sarcopenia ^§^	1.357	0.244	1.143	0.63
	(0.812–2.269)		(.663–1.971)	

Abbreviations: OR, odds ratio; CI, confidence interval; HGS, hand grip strength; ASMI, appendicular skeletal muscle mass index; SPPB, short physical performance battery. ^†^ Low HGS (<28 kg for men and <18 kg for women); Low ASMI, <7.0 kg/m for men and <5.4 kg/m for women; Low SPPB ≤ 9 for both sexes; ^††^ Sarcopenia: low HGS and low ASMI. ^§^ Severe sarcopenia, low HGS, low ASMI, and low SPPB. The fully adjusted model was adjusted for age, sex, depression, osteoarthritis, osteoporosis, diabetes mellitus, hypertension, smoking, alcohol consumption, location of residence, and body mass index. * *p* < 0.05.

## Data Availability

This study used data from the KFACS database. Data provision manuals and contact information are available at the KFACS website (http://www.kfacs.kr, accessed on 25 November 2021).

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
