# Peer review of "Impact of Vitamin B12 Insufficiency on Sarcopenia in Community-Dwelling Older Korean Adults"

_ijerph, 2021, doi:10.3390/ijerph182312433_

Round 1
Reviewer 1 Report
The manuscript of Chae et al. focuses on the role of vitamin B12 deficiency on sarcopenia severity by a cross-sectional study of a large cohort of old sarcopenic subjects. Authors analyzed muscle mass indexes and muscle function after dividing the selected subjects in two groups based on serum levels of vitamin B12. Statistical analyses show that vitamin B12 insufficiency significantly associates with a lower index of appendicular muscle mass, but not with severe loss of muscle function or severe sarcopenia. The findings are interesting. Some aspects, however, need to be considered and explained further
Major concern
- Authors’ definition that “sarcopenia directly causes a decrease in muscle strenght” (line 44) is questionable. Sarcopenia has been defined recently by the European Working Group on Sarcopenia of Older People (2019) as the simultaneous loss of muscle mass and force. In fact, the isolated loss of muscle mass (muscle atrophy) may not be followed by loss of force and has been dubbed pre-sarcopenia. Authors should clearly state here if enrolment excluded subjects displaying muscle mass loss without muscle function loss, not in the Materials and Methods section. They should also quote literature showing that, in humans, the loss of muscle force may be more severe than the loss of muscle mass, as the rationale for the severe sarcopenia group.
- The list of “factors related to sarcopenia” needs additional clarifications (line 52). What does it means “reducing muscle cells”? Insulin-resistance is not an obligatory feature of sarcopenia, whereas it accompanies other types of muscle atrophy, such as muscle disuse atrophy and cachexia. Please clarify.
- Macrocitic anemia has not been quoted among the consequences of vitamin B12 deficiency. However, all the subjects enrolled show borderline levels of hemoglobin. What about red blood cell number and volume? Would this parameter be helpful in reconsidering the minimum level of vitamin B12, which was chosen as the requirement for myelin synthesis? Would it lead to divide subjects in three groups instead of two? The levels of vitamin B12 required to maintain proliferative ability of cell precursors might be more relevant for sarcopenia, than those for myelin (see comment 4).
- Authors did not consider in their discussion the consequences of vitamin B12 deficiency on muscle precursors (satellite cells), whose reduced ability to proliferate and be committed is presently considered as one major factor responsible for age-related muscle atrophy development (see McCormick and Vasilaki, 2018)
Minor concers
- Typos (dash within words) in the Abstract
- Table 1. Table readability is not immediate. Please add: n (%), when indicating number of subjects
- Please, be realistic about the biological significance (about 6cm/sec) of the statistical significant difference between the values of 4m walking speed of the two B12 groups.
Reviewer 2 Report
I commend the authors in attempting to contribute to this body of literature regarding the sarcopenia reasons in the elderly population. Seon A Chae et al. provides information concerning the “Impact of Vitamin B12 Insufficiency on Sarcopenia in Community-dwelling Older Adults”.
The relevance of this topic in the risk of sarcopenia is indisputable. Therefore, I recommend that this manuscript be accepted. Moreover, some corrections to improve the transparency and communicativeness of the study are needed.
- Study flow chart of the participant recruitment process has very weak resolution, maybe it is since system transformation please check it
- On the flowchart “Excluded if…” it seems to be confusing, please correct for Exclusion criteria and list them, showing how many subjects were rejected since of them.
- In the whole text there are some editorial gaps like “pe-ripheral” – should be “peripheral” (this is the only example needed to review the whole text.
- In spite of the quite large number of subjects from KFACS there is a lack of information about minimum sample size calculation. It should be delivered as an explanation representativeness of the analyzed sample.
- My doubts concerning the conclusion where Authors generalize the reasons for sarcopenia without mentioning that the study sample consisted only of elderly subjects.
- This group is usually affected by civilization diseases, and the treatment of different medicines, possibly related to muscle mass changes. Authors should discuss this aspect in results and conclusions as well.
Round 2
Reviewer 1 Report
Response to Point 1 (Page 1, line 44-45): Defintition of sarcopenia is still unacceptable. Please, correct as follows:
Sarcopenia occurs when the age-related loss of skeletal muscle mass is accompanied by muscle strength loss or physical dysfunction. This condition is responsible for disability and …..
Response to point 2 (page 2, lines 52-55): The new formulation of the sentence is misleading. Muscle disuse and cachexia can not be considered as factors leading to primary sarcopenia development, despite of the existence of similarities among the involved mechanisms. Especially the consideration of the latter one is puzzling, since primary sarcopenia affects healthy aged subjects. I wonder whether subjects affected with cancer were enrolled in the study (they would be affected with secondary sarcopenia). Please clarify.
Current knowledge recognizes age-related loss of muscle innervation and reduced myonuclei replacement by satellite cells among the major factors leading to sarcopenia development. Please, read and quote Larsson et al. Sarcopenia: Aging-Related Loss of Muscle Mass and Function. Physiol Rev. 2019, 99, 427–511. This should be mentioned, because vitamin B12 involvement in myelin synthesis and muscle precursor proliferation might attenuate these two sarcopenia-inducing mechanisms.
Response to point 4 (page 7, line 202-204): The Authors' definition of satellite cells as muscle stem cells necessary for muscle regeneration is only partially true. As explained in the previous comment, satellite cells provide myonuclei to adult myofibers too.
